# Integration of Rap1 and Calcium Signaling

**DOI:** 10.3390/ijms21051616

**Published:** 2020-02-27

**Authors:** Ramoji Kosuru, Magdalena Chrzanowska

**Affiliations:** 1Versiti Blood Research Institute, Milwaukee, WI 53201, USA; rkosuru@versiti.org; 2Department of Pharmacology and Toxicology, Medical College of Wisconsin, PO Box 2178, Milwaukee, WI 53201-2178, USA; 3Cardiovascular Center, Medical College of Wisconsin, PO Box 2178, Milwaukee, WI 53201-2178, USA

**Keywords:** Rap1, calcium, CalDAG-GEF, Epac, SERCA

## Abstract

Ca^2+^ is a universal intracellular signal. The modulation of cytoplasmic Ca^2+^ concentration regulates a plethora of cellular processes, such as: synaptic plasticity, neuronal survival, chemotaxis of immune cells, platelet aggregation, vasodilation, and cardiac excitation–contraction coupling. Rap1 GTPases are ubiquitously expressed binary switches that alternate between active and inactive states and are regulated by diverse families of guanine nucleotide exchange factors (GEFs) and GTPase-activating proteins (GAPs). Active Rap1 couples extracellular stimulation with intracellular signaling through secondary messengers—cyclic adenosine monophosphate (cAMP), Ca^2+^, and diacylglycerol (DAG). Much evidence indicates that Rap1 signaling intersects with Ca^2+^ signaling pathways to control the important cellular functions of platelet activation or neuronal plasticity. Rap1 acts as an effector of Ca^2+^ signaling when activated by mechanisms involving Ca^2+^ and DAG-activated (CalDAG-) GEFs. Conversely, activated by other GEFs, such as cAMP-dependent GEF Epac, Rap1 controls cytoplasmic Ca^2+^ levels. It does so by regulating the activity of Ca^2+^ signaling proteins such as sarcoendoplasmic reticulum Ca^2+^-ATPase (SERCA). In this review, we focus on the physiological significance of the links between Rap1 and Ca^2+^ signaling and emphasize the molecular interactions that may offer new targets for the therapy of Alzheimer’s disease, hypertension, and atherosclerosis, among other diseases.

## 1. Discovery, Early and Classical Functions of Rap1: Ras Antagonism, Integrin Activation

Rap1, a 21 kDa monomeric G-protein, was discovered in 1989 by Noda and his coworkers in a screen for proteins able to suppress the oncogenic effect of *K-Ras* (one of the mutated Ras genes) [1]. Described as Kristen-ras-revertant-1 (Krev-1), the protein was found to have high similarity to Ras proteins [2]. Simultaneously, Pizen et al. characterized two proteins, Rap1 and Rap2, as Ras homologues and proposed that Rap1, identical to Krev-1, might function as an antagonist of Ras by competing for a common target, or mediating growth inhibitory signals independently of Ras [3,4]. Since then, many groups have reported that Rap1 antagonizes Ras signaling by trapping its effector proteins, serine/threonine kinase Raf, in an inactive complex [5]. However, much research has also demonstrated the functions of Rap1 independent of Ras.

The two highly conserved Rap1 isoforms, Rap1a and Rap1b, share 95% sequence identity, with a 50% sequence homology to Ras [3]. The basic structure of Rap1 is similar to Ras and consists of a catalytic domain made of a six-stranded central β-sheet (β1–β6) surrounded by five α-helices (α1–α5) and ten loops (L1–L10) [6,7]. The two regions of highest sequence similarity between Ras and Rap1 correspond to the switch 1 (amino acids 32–38) and switch 2 (amino acids 60–70) regions [7,8]. These regions adopt different conformations when bound to GTP (active) or GDP (inactive) and allow effector proteins to discriminate between the active and inactive form of small G protein. Despite the identical effector domains and a shared subset of effectors, many of Rap1’s biological functions are distinct from Ras, due to cellular and signaling differences in the utilization of the same effectors [9]. Furthermore, Rap1 controls cell adhesion by modulating the activity of adhesion receptors—integrins and cadherins—through specific interactions with its effectors: RAPL, Riam, AF-6, Krit1, Vav2, Tiam1, and Arap3, [9,10,11].

The kinetics of the GDP–GTP cycle is governed by diverse families of guanine exchange factors (GEFs) containing a Ras exchange motif (REM), a catalytic Cdc25 homology domain with nucleotide exchange activity, and additional regulatory domains which enable a wide variety of regulatory mechanisms (Table 1) [9,12]. Two of those families—CalDAG-GEFs, activated by Ca^2+^ and diacylglycerol (DAG), and Epac proteins, activated by cyclic adenosine monophosphate (cAMP)—are of particular importance for coordinating Rap1 and Ca^2+^ cross talk, and will be discussed in more detail in the following sections. In addition to regulation by GEFs, Rap1 undergoes a series of posttranslational modifications that determine its activity and cellular functions.

## 2. Posttranslational Modifications and Cellular Localization of Rap1

Rap1 is a soluble cytosolic protein that undergoes isoprenylation (geranylgeranylation), a covalent binding of geranylgeraniol to the –SH group of cysteine in the C-terminal CAAX motif (Cys-aliphatic residue-aliphatic residue-X amino acid sequence; X- usually Met, Gln, Ser or Leu). This posttranslational modification, combined with CAAX motif cystine carboxymethylation, enhances Rap1 hydrophobicity and facilitates its membrane localization [13,14,15]. Rap1 can also be modified by phosphorylation, which provides another layer of functional regulation, and is catalyzed by protein kinase A (PKA) [16,17]. PKA-mediated phosphorylation at serine 180 and serine 179 positions on Rap1a and Rap1b, respectively, promotes the direct binding of Rap1 to scaffold protein KSR (kinase suppressor of Ras) and enables the coupling of B-Raf to extracellular signal-regulated kinase-1 (ERK) and sustained ERK activation. Consequently, the phosphorylation of Rap1 has been implicated in the regulation of cell differentiation and growth [17,18].

Signaling events that interfere with the prenylation of Rap1 may decrease its membrane localization and, thus, interfere with cell–cell adhesion. For instance, adenosine A2B receptor-mediated signaling induces Rap1b (ser179 and ser180) phosphorylation and leads to decreased binding to chaperone protein small G-protein dissociation stimulator (SmgGDS). This signaling inhibits Rap1 prenylation and membrane localization and results in cell scattering [19,20].

Cyclic nucleotide phosphodiesterases (PDEs) are key regulators of cAMP signaling. A component of scaffolding complexes that contain A-kinase anchoring proteins such as: PKA, Epac, and adenylate cyclase [21], PDEs directly interact with prenylated Rap1 to control its function in various cell types. In vascular endothelial cells, interaction of PDE4D and Epac1 is critical for the integration into the VE-cadherin-based signaling complex and the coordination of cAMP-mediated vascular endothelial cell adhesion and permeability [22]. PDE6δ (retinal rod rhodopsin-sensitive cGMP 3′,5′-cyclic phosphodiesterase, subunit delta) interacts with prenylated Rap1 in neurons and interferes with its trafficking, thereby dissociating it from the cell membrane, which is where Rap1 promotes Ca^2+^ influx [23]. In this way, the inhibition of Rap1 interaction with PDE6δ has been proven to be beneficial in restraining disease-associated, abnormal Ca^2+^ influx and neuronal hyperactivity, and providing neuroprotection in models of Alzheimer’s disease [24].

In addition to localizing at the plasma membrane [25], Rap1 is present at other membranes, including Golgi apparatus and late endocytic compartments [26,27]. Specifically, the subcellular localization of Rap1 pools determines Rap1 coupling to its effectors and its susceptibility to GEF regulation. This has key functional significance for Rap1-regulated processes such as exocytosis [28,29] and integrin-mediated adhesion. Epac1 activity towards Rap1 depends on Rap1 subcellular localization; in effect, Epac activates the plasma membrane, but not perinuclear pools of Rap1. The activation of plasma membrane-localized Rap1 promotes ERK activation and granule secretion [30]. Ca^2+^ and DAG-dependent Rap1 GEF, CalDAG-GEFI, localization at plasma membrane is key for Rap1 activation and, subsequently, integrin activation in platelets [31]. All these Rap1 regulatory factors contribute to Rap1 and Ca^2+^ signaling cross-talk.

## 3. Ca^2+^ Signaling and Rap1

Ca^2+^ is a ubiquitous secondary messenger, responsible for controlling a myriad of key cell processes, including fertilization, proliferation, contraction, and neural signaling and learning, such as synaptic plasticity, neuronal survival, chemotaxis of immune cells, platelet aggregation, vasodilation, and cardiac excitation-contraction coupling [32,33,34,35,36]. In most cells under basal conditions, the cytosolic concentration of free Ca^2+^ is approximately 100 nM, which is 10,000 times less than that of extracellular Ca^2+^. Upon stimulation, intracellular Ca^2+^ levels rapidly, but transiently, rise to above 1 μM. However, sustained increases in intracellular free Ca^2+^ to the micromolar range are deleterious to cellular functions and the efficient lowering of intracellular Ca^2+^ via Ca^2+^ buffers and uptake by Ca^2+^ pumps is essential for preventing cell damage or death.

The rise in cytoplasmic Ca^2+^ levels can be generated either from intracellular stores or extracellular sources. Ca^2+^ release from internal stores is controlled by several channels, of which the inositol 1,4,5-trisphosphate (IP_3_) receptor-mediated Ca^2+^ release from endoplasmic reticulum is a universal and highly versatile mechanism [37,38]. Most of the drugs/agonists that act on the G protein-coupled receptor (GPCR) and tyrosine kinase receptors (TKR) utilize the IP_3_-mediated Ca^2+^ release pathway to promote their signal transduction [39]. Phospholipase C (PLC) activated downstream from GPCRs and TKRs, cleaves phosphatidylinositol 4,5 bisphosphate into IP_3_ and diacylglycerol (DAG). Liberated IP_3_ then binds to IP_3_ receptors present on endoplasmic reticulum to allow Ca^2+^ release, and increases cytoplasmic Ca^2+^ levels [39]. On the other hand, extracellular Ca^2+^ entry is regulated by several channels that open after the depletion of intracellular Ca^2+^, which include voltage-gated Ca^2+^ channels, receptor-operated Ca^2+^ channels and store-operated Ca^2+^ channels [40]. The combined action of intracellular Ca^2+^ release and extracellular Ca^2+^ entry is required to tightly control changes in the length and amplitude of Ca^2+^ fluxes to regulate multiple signaling pathways [37,38].

The universal mechanisms that lead to Ca^2+^ release also generate signals that activate Rap1. Multiple modalities of Rap1 GEF activation allow Rap1 to act as an effector, as well as an upstream regulator of Ca^2+^ signaling. Downstream from Ca^2+^ and DAG generated in parallel to the induction of the Ca^2+^ signal - Rap1 acts as one of Ca^2+^ signaling effectors. Activated by other GEFs, in particular by Epac in response to elevated cAMP, Rap1 controls Ca^2+^ signals. How these pathways intersect to exert multiple, tissue-specific effects is described in more detail below.

## 4. Rap1 Activators in Integration of Ca^2+^ Signaling

Rap1 signaling is remarkably complex, with cross-talk between multiple receptors and its interacting effector proteins [16,41,42,43]. Rap1 activity is controlled by several evolutionarily conserved families of GEFs, and, in particular, Ca^2+^ and DAG-activated CalDAG-GEFs and 3′ and 5′-cyclic adenosine monophosphate (cAMP)-activated Epacs. These two GEF families are of key importance for the cross-talk between Rap1 and Ca^2+^ signaling.

### 4.1. CalDAG-GEFs

The discovery of a Ca^2+^-binding GEF, CalDAG-GEFII, encoded by *RASGRP1* gene, with an activity towards Ras, introduced an intriguing possibility of a cross-talk between the Ca^2+^ and Rap1 signaling pathways [44]. This possibility materialized when a second family member, CalDAG-GEFI (*RASGRP2*) was identified as a novel brain transcript, and was shown to activate Rap proteins [45,46]. Subsequently, other family members: CalDAG-GEFIII (*RASGRP3*) and CalDAG-GEFIV (*RASGRP4*) were identified as regulators of various Ras proteins (Table 1) in B-cells and mast cells [47,48]. Different CalDAG-GEF isoforms are present in most tissues, including the hematopoietic and neuronal cells where some of their functions have been characterized, and in blood vessels [45,49,50,51,52,53,54].

The four CalDAG-GEF family members (I, II, III and IV) share a similar structure containing conserved Cdc25 homology domain (catalytic site), a Ras exchange motif (REM), and two atypical EF hands involved in Ca^2+^ binding and release of autoinhibition involved in GEF activation (in case of CalDAG-GEFI) [55]. In addition, a C-terminal C1 motif that mediates lipid interactions is important for the localization and/or activation of CalDAG-GEFs. Except for CalDAG-GEFI, which contains atypical C1, the remaining CalDAG-GEFs contain typical C1 motifs with a high affinity for DAG. The differences in C1 domains contribute to the differential regulation of CalDAG-GEFs by Ca^2+^ and DAG [56,57,58] (Table 1). Both CalDAG-GEFII and CalDAG-GEFIII contain typical C1 domains with high affinity for DAG and translocate to the plasma membrane after treatment with DAG mimetic 12,13-tetradecanoyl phorbol acetate (TPA), but are insensitive to increased levels of Ca^2+^ [59,60,61,62]. In contrast, the atypical C1 domain of CalDAG-GEFI has low affinity for DAG [63], but high affinity for plasma membrane phosphoinositides PIP_2_ and PIP_3_. This atypical C1 domain is required for CalDAG-GEFI association with the plasma membrane [31].

The four CalDAG-GEF family members exhibit different GTPase specificities depending on the availability of Ca^2+^ and DAG (Table 1) [12,60]. For example, CalDAG-GEFI functions as a dual R-Ras/Rap1 activator and Ca^2+^ regulation plays a key role in determining its specificity. Since Ca^2+^ stimulates the Rap-exchange activity of CalDAG-GEFI, while inhibiting the Ras-exchange activity, a cytosolic Ca^2+^ signal effectively shifts the catalytic activity of CalDAG-GEFI from Rap to Ras GTPases [45,46]. CalDAG-GEFII specifically functions as a Ras and R-Ras activator while CalDAG-GEFIII activates several Ras GTPases, including Rap1, Rap2, Ras, and R-Ras [44,45,60]. Two CalDAG-GEFs act via Rap1 and the structural difference in the C1 domain determines the mechanism of their activation and signaling context. CalDAG-GEFI promotes Rap1 activation via Ca^2+^ while CalDAG-GEFIII mediates Rap1 activation via DAG. CalDAG-GEFI-Rap1 signaling is important in central nervous system (CNS) and platelet function. While the exact functions of DAG-activated CalDAG-GEFIII are less well understood, it is important in macrophage activation and has been linked with hypertension through GWAS studies [53].

### 4.2. Epac

Rap1 is an important mediator of cellular cAMP signaling [64]. Elevation in cAMP levels, resulting from adenylyl cyclase activation downstream from ligand-induced G_αs_-coupled GPCRs stimulation, induces the activation of Rap1 GEFs and Epacs (exchange proteins directly activated by cAMP) [64,65,66]. Two members of Epac family, Epac1 and Epac2, catalyze the guanine nucleotide exchange on Ras GTPases, including Rap1 and Rap2 [66,67]. Epac1 is ubiquitously expressed in the CNS, heart, and other organs, including the kidney, spleen, pancreas, ovary, thyroid, adrenal glands, as well as the endothelium. Epac2 is predominantly expressed in the brain and the adrenal glands [66,67].

Epac1 and Epac2 share a similar structural organization, with the C-terminal catalytic GEF region and N-terminal regulatory region. The catalytic regions of Epacs possess a Ras exchange motif (REM domain), a Cdc25-homology catalytic domain that mediates the GEF activity for Rap GTPases and a RAS-association domain (RA domain), which translocates Epac2 to the plasma membrane. The regulatory region of Epac1 consists of a DEP domain (Dishevelled, Egl-10, and Pleckstrin) that is responsible for membrane anchoring, and a conserved cAMP-binding domain [66,68] (Table 1). In the unbound state, the cAMP-binding domain acts as an auto-inhibitory module for the catalytic Cdc25-homology domain. The binding of cAMP induces conformational changes in hinge helix and allows the regulatory region to move away from the catalytic region, thereby exposing the GEF domain to allow Rap1 binding [69]. Although they are similar in domain structure, Epac2 differs from Epac1 in the additional N-terminal cAMP-binding domain, which binds cAMP with a much lower affinity and is unable to induce GEF activity after cAMP binding [69,70].

Epac plays an important role in the regulation of Ca^2+^ homeostasis and Epac and Ca^2+^ signaling pathways crosstalk at multiple levels, converging on effectors like IP_3_ receptor and ryanodine receptor (RyR), mediating Ca^2+^ release, or sarcoendoplasmic reticulum Ca^2+^-ATPases (SERCA), mediating Ca^2+^ clearance, effectively forming a signaling network in non-excitable cells [71,72,73]. The signaling schemes include Epac acting as an inducer of Ca^2+^-induced Ca^2+^ release to mobilize intracellular Ca^2+^ levels, as found in the regulation of exocytosis in human pancreatic β-cells and INS-1 insulin-secreting cells [74,75]. Interestingly, Ca^2+^ can also modulate the Epac signaling pathway by activating the adenylyl cyclase to increase the production of cAMP levels [76]. It is important to note that downstream effects are dependent on the distinctive activation of subcellular pools of Rap1. While not all of Epac functions are mediated by Rap1 [30,64], the Epac/Rap1 axis and Ca^2+^ signaling intersect to regulate important functions in several tissues. Some of the mechanisms uncovered are described below.

## 5. Integration of Rap1 and Ca^2+^ Signaling in the Central Nervous System (CNS)

Via the activation of the ERK signaling pathway, Rap1 is involved in a number of Ca^2+^-dependent processes, such as neuronal excitability, synaptic plasticity, long-term potentiation and gene transcription [77,78,79,80]. In most cases, the activation of Rap1, and the ensuing B-Raf1 and ERK activation, depends on agonist-induced Ca^2+^ influx leading to CalDAG-GEFI activation and the subsequent formation of CalDAG-GEFI/Rap1/B-Raf cassette to stimulate the ERK pathway [81] (Figure 1). The mechanisms that lead to Ca^2+^-dependent Rap1 activation vary in different neuronal cells depending on the source and magnitude of Ca^2+^ signal. In PC12 and hippocampal neurons, Rap1-ERK signaling is mediated by PKA activation upon depolarization-induced Ca^2+^ influx through L-type Ca^2+^ [79] (Figure 1). In primary striatal neurons, the dopamine D1 receptor-induced, PKA-mediated Ca^2+^ release activates the Rap1/B-Raf/ERK pathway to regulate cAMP-response element binding protein (CREB)-phosphorylation and gene expression [77] (Figure 1). This mechanism of Rap1 activation appears to be primed by PKA-induced Ca^2+^ release, but is not further induced by direct or indirect PKA- or protein kinase C-dependent phosphorylation [77]. Thus, Ca^2+^ and Rap1 signaling can intersect at multiple signaling modules.

Furthermore, the magnitude of intracellular Ca^2+^ activates different pools of Rap1 to mediate ERK signaling at spatially discrete, subcellular locations, which is essential for controlling spatially discrete processes underlying neuronal function and survival. For example, in resting neurons, steady-state levels of Ca^2+^ and cAMP drive the activation of the membrane-associated pool of Rap1-ERK signaling [78], which leads to a reduction in the A-type K^+^ channel Kv4.2 activity that controls the back-propagation of action potentials (or action potential repolarization) in hippocampal CA1 pyramidal neurons [82]. Conversely, in depolarizing neurons, Ca^2+^ influx stimulates the nuclear pool of Rap1-ERK signaling, which phosphorylates nuclear targets involved in the expression of hippocampal long term potentiation [78] and CREB-dependent gene transcription [83]. Thus, in these cellular scenarios, Rap1 is a mediator of Ca^2+^-signaling.

In addition to acting as an important mediator of Ca^2+^-mediated ERK signaling, Epac-mediated Rap1 activation controls Ca^2+^-dependent signaling events, such as resting membrane potential, glutamate release, and cortico-amygdala plasticity in neurons. In mouse cerebellar granule cells, the Epac-induced activation of Rap1 and p38 mitogen-activated protein kinase (MAPK) mobilizes intracellular Ca^2+^ release, facilitating the opening of large conductance Ca^2+^-activated K^+^ channels to modulate resting membrane potential and after-hyperpolarization [84]. In primary cortical neurons ERK1/2 and L-type, Ca^2+^ channels act as the downstream Rap1 effectors to mediate the suppression of glutamate release required for cortico-amygdala plasticity and fear learning. Rap1 deletion in these cells leads to increased axonal Ca^2+^ influx, ERK inhibition, and increased plasma membrane expression of L-type voltage-gated Ca^2+^ channels (Ca_v_1.2 or Ca_v_1.3), enabling the Ca^2+^-regulated glutamate release [80].

Interestingly, the modulation of prenylated Rap1 appears to play a role in controlling disease-associated Ca^2+^ aberrations and neuronal activity. The inhibition of Rap1 interaction with PDE6δ restrains disease-associated abnormal Ca^2+^ influx and neuronal hyperactivity and confers neuroprotection in models of Alzheimer’s disease [24]. Rap1 may be an important therapeutic target for the treatment of neuro-degenerative disorders associated with Ca^2+^ aberrations, such as Alzheimer’s disease.

## 6. Platelets: Integrins and SERCA

Early studies demonstrated that an increase in cytosolic Ca^2+^ is required and sufficient for Rap1 activation in platelets [41]. Much research has implicated the CalDAG-GEFI–Rap1 signaling axis as a mediator of that activation [85] and recent studies revealed a Ca^2+^-dependent mechanism of CalDAG-GEFI, and downstream, Rap1 activation. In resting platelets, where cytosolic Ca^2+^ levels are low, CalDAG-GEFI remains in an auto-inhibited state. In response to agonist stimulation, elevated cytosolic Ca^2+^ binds to the EF hands and induces structural rearrangements that free the catalytic surface of CalDAG-GEFI to activate Rap1b [55]. Downstream from activated Rap1, its effector, Rap1-GTP-interacting adaptor molecule (RIAM), recruits talin to β_3_ integrin subunit and contributes to integrin activation [86,87]. While the functional significance of RIAM in α_IIb_β_3_ integrin activation has not been fully validated in vivo [88,89], evidence points to CalDAG-GEFI’s importance in hemostasis. CalDAG-GEFI deficiency in platelets leads to delayed Ca^2+^-dependent rapid activation of Rap1 and a marked defect in platelet aggregation [49], similar to the phenotype of Rap1b knockout mice [90]. Human platelets expressing an inactive CalDAG-GEFI are defective at clot formation, which points to a fundamental role for CalDAG-GEFI–Rap1 signaling module in platelet Ca^2+^ homeostasis [91]. Importantly, Rap1 activation by CalDAG-GEFI is a critical signaling step linking Ca^2+^ signaling with integrin α_IIb_β_3_ activation, thromboxane A2 formation, and granule release in platelets [49,92,93] (Figure 2).

In addition to acting as an effector of Ca^2+^-induced CalDAG-GEFI in platelet integrin activation, Rap1 has been implicated in the regulation of IP_3_-sensitive intracellular Ca^2+^ pools via the regulation of sarcoendoplasmic reticulum Ca^2+^-ATPases (SERCA) [94,95]. SERCA is a key Ca^2+^ pump that transports Ca^2+^ into ER, reducing cytosolic Ca^2+^ concentration, thereby controlling IP_3_-sensitive intracellular Ca^2+^ pools in platelets [96]. Studying pathological platelets obtained from congestive heart failure patients, Magnier et al., observed a decrease in the expression and phosphorylation of Rap1 that correlated with the reduced expression of 97 kDa SERCA in the platelets of congestive heart failure patients [94]. Later, Lacabaratz-Porret et al., demonstrated that this 97 kDa SERCA, a SERCA3b isoform, physically interacts with Rap1b protein, which suggests that SERCA 3b is a target of Rap1b [95].

The dynamic regulation of the interaction between Rap1b and SERCA 3b by cAMP-dependent phosphorylation of Rap1 may act to regulate a transition between platelet inhibition and activation [97,98]. Increased cAMP production leads to the phosphorylation of Rap1b and its subsequent dissociation from SERCA 3b protein results in the stimulation of its activity to enhance the filling state of SERCA-associated Ca^2+^ pool to induce platelet inhibition [95] (Figure 2). In diseased or hypertensive platelets, decreased cAMP leads to a decrease in phosphorylation of Rap1b and SERCA 3b activity. This results in a smaller SERCA-associated Ca^2+^ pool, thus decreasing IP_3_-sensitive Ca^2+^ release to promote platelet activation [95]. Thus, the interplay between SERCA 3b and Rap1-modulating phosphorylation may be clinically significant in cardiovascular pathology.

## 7. Rap1 and Ca^2+^ Signaling in the Immune System: TLR, Integrins, and Chemotaxis

Similarly to platelets, the activation of integrins is one of the best characterized Rap1 functions in leukocytes that intersects with Ca^2+^ signaling, with CalDAG-GEFI acting as Rap1 activator [99,100] and RIAM and talin as Rap1 effectors [101,102]. Integrin activation is fundamental for leukocyte migration, chemotaxis and trafficking, and cell adhesion [103,104].

CalDAG-GEFI deficiency in neutrophils impairs F-actin formation, E-selectin-dependent slow rolling, adhesion, and speed and the directionality of migration [100,105,106]. In vivo, CalDAG-GEF1 deficiency blocks TNFα-induced intravascular neutrophil adhesion and recruitment during sterile peritonitis [105]. Decreased CalDAG-GEF1/Rap1 signaling, as in cases of genetic deletion of *RASGRP2* in mice [100] or loss of function mutations in humans [107,108], has been suggested as a cause of the rare leukocyte adhesion deficiency type III (LAD-III). However, mutations in Kindlin-3 were found to be causative of LAD-III [109,110], solving that controversy [111]. On the other hand, increased CalDAG-GEFI/Rap1 signaling is responsible for increased cell migration in chronic lymphocytic leukemia (CLL) downstream from acyclic ADP ribose hydrolase, CD38 [112]. Elevated expression of CD38 leads to elevated intracellular Ca^2+^ [113] and activates Rap1 via CalDAG-GEFI, subsequently leading to activation of integrin and facilitating CLL adhesion [112].

Once activated by CalDAG-GEFI or Epac, Rap1 controls chemotaxis and the trafficking of immune cells via additional, distinct mechanisms. In neutrophils, CalDAG-GEFI-activated Rap1 controls chemotaxis in an integrin-independent manner through a mechanism that involves actin cytoskeleton and cellular polarization [106]. In lymphokine-activated killer (LAK) cells, Epac–Rap1 activation downstream from endoplasmic reticulum Ca^2+^ release triggers the production of nicotinic acid adenine dinucleotide phosphate (NAADP) and enables Ca^2+^ release from lysosomal acidic organelles to stimulate long-lasting Ca^2+^ entry through transient receptor potential melastatin 2 (TRPM2) channels required for cell migration [114] (Figure 3).

In addition to the regulation of migration, Rap1 signaling is an important modulator of Ca^2+^-dependent regulation of toll-like receptor (TLR) signaling in immune cells [115,116]. The intensity of pathogenic TLR stimuli and the corresponding intensity of Ca^2+^ signal has been linked with differential activation signaling by Ras and Rap1 as well as a differential effect on ERK activation and cytokine production [116,117]. Induced by low-intensity TLR stimuli, low-intensity Ca^2+^ influx mediated by stromal interaction molecule 1 (STIM1) favors Rap1 inhibition and ERK activation, while high-intensity TLR stimuli trigger more intense Ca^2+^ influx, leading to Ras activation and cytokine production [116] (Figure 3). Interestingly, this effect is mediated by CalDAG-GEFIII, which limits TLR-mediated cytokine production by activating Rap1 and ERK in response to a low level of antagonists and, in vivo, limits the inflammatory response [117]. These findings underscore the importance of Rap1 in the modulation of the Ca^2+^-dependent, key aspects of the immune response.

## 8. Heart: Excitation-Contraction Coupling; Cardiac Hypertrophy

Epac proteins play an important role in the regulation of cardiac physiology, with some of its functions being mediated by Ras family members other than Rap1 [64,118,119]. Nonetheless, the Epac–Rap axis intersects with cardiac Ca^2+^ signaling to regulate cardiac excitation–contraction coupling, with phosplipase Cε (PLCε) and CamKII as Epac effectors [64,71]. PLCε acts as an effector of both Epac and Rap GTPase Rap2b [120]. Importantly, by activating Rap via its Rap GEF activity, PLCε facilitates Ca^2+^-induced Ca^2+^ release (CICR) in adult ventricular cardiomyocytes. In response to β-adrenergic receptor (βAR) stimulation, Epac–Rap2b induce both Rap GEF activity and hydrolytic activity of PLCε. This leads to sustained Rap activation and the induction of PKCε and CamKII phosphorylation downstream from PIP_2_ breakdown. These signaling events lead to enhanced sarcoplasmic reticulum Ca^2+^-induced Ca^2+^ release (CICR) [121,122] (Figure 4). Thus, Epac–Rap modulates cardiac Ca^2+^ homeostasis through the regulation of CICR in cardiomyocytes. Although these studies identified Rap2b as the main isoform involved in the cardiac Ca^2+^ signaling, the role of Rap1 in this context is not known. Therefore, Rap1 isoform-specific knockout animal models will be useful to determine the in vivo role of Rap1 in cardiac Ca^2+^ signaling.

In addition to its role in cardiac physiology, Epac is also implicated in cardiac pathology. Epac stimulation induces cardiac hypertrophy via activation of IP_3_-induced intracellular Ca^2+^ rise, leading to the activation of the numerous Ca^2+^ sensitive hypertrophic proteins, including calcineurin, histone deacetylases, and nuclear factor of activated T cells (NFAT) [123,124]. In cardiomyocytes, Epac proteins function as signalosomes [64]—macromolecular complexes which consist of mAKAP (muscle A kinase-anchoring protein), protein kinase A, phosphodiesterase PDE4D3, ryanodine receptor, phosphatases PP2A, and calcineurin, and serve as signaling nodes in the Ca^2+^ signaling network [125]. While the significance of Epac for Ca^2+^ homeostasis and heart function is indisputable, the identity and function of its specific effectors, including Rap GTPases, in normal and pathologic conditions, remain to be fully elucidated.

## 9. Vascular Smooth Muscle Cells: Vasorelaxation

Rap1 plays an important role in the regulation of vascular tone. At least two distinct mechanisms connect Ca^2+^ and Epac-dependent Rap1 activation in vascular smooth muscle cells to control smooth muscle relaxation. The activation of Epac modulates Ca^2+^ sensitivity of the contractile proteins by a Rap1-dependent reduction in RhoA GTPase activity in several types of smooth muscle cell from airway, gut, and vascular tissues [126,127]. In these cells, Rap1 activation downstream of Epac leads to reduced RhoA activity. This induces a series of events, including decreased myosin regulatory light chain (RLC_20_) phosphorylation and the disinhibition of myosin light chain phosphatase (MLCP) activity, which leads to Ca^2+^-desensitization and relaxation of force in smooth muscle [126,127] (Figure 4). Consistently, Epac-induced vasorelaxation is decreased in Rap1b knockout vascular smooth muscle cells through inhibition of RhoA-mediated sensitization to Ca^2+^, medicated by decreased RLC_20_ phosphorylation [128]. Importantly, Rap1b deficiency led to the development of hypertension, in part via functional changes to vascular smooth muscle cells [128].

In addition to triggering signaling that modulates Ca^2+^ sensitivity of contractile proteins, Epac–Rap1 may act directly to regulate Ca^2+^ influx, thus inducing hyperpolarization of smooth muscle membrane and leading to vasorelaxation. Epac activation increases the activity of Ca^2+^ sparks from ryanodine receptors to open Ca^2+^-sensitive K^+^ channels (BK_Ca_), inducing smooth muscle hyperpolarization. This, subsequently, leads to a decrease in the activity of voltage-gated Ca^2+^ channels, reducing Ca^2+^ influx and promoting vasorelaxation [129] (Figure 4). Altogether, these studies indicate that Rap1, by altering Ca^2+^ sensitivity of vascular smooth muscle cells, plays an important role in maintaining normal vascular contractile state and contributes to blood pressure regulation.

## 10. Endothelium: NO, Vasorelaxation, Vasoreactivity

Some of the best-described functions of Rap1 in the endothelium involve the dynamic regulation of endothelial junctions and the control of the endothelial and vascular barrier [11,130]. Acting via its effectors Rasip, Radil, and afadin (AF6), Rap1 facilitates interactions between adherens and tight junction components and, by regulating the activity of Rho small GTPases, orchestrates actin cytoskeletal rearrangements to enhance endothelial barrier [131,132,133,134,135]. While multiple Rap1 GEFs are involved in the dynamic regulation of the endothelial barrier [130], cAMP/Epac -activated Rap1 plays a particularly important role in lung vasculature in vivo by protecting against ventilator- or inflammation-induced lung injury [136,137]. In pulmonary endothelium in vitro, Rap1 geranylgeranylation has been linked with Ang II-mediated activation of Ca^2+^-activated tyrosine kinase Pyk2. This finding suggests the involvement of functional Rap1 in hypertension and vascular permeability [138]. However, the physiological significance of the Rap1 activation by AngII pathway is unknown [139,140]. While Ca^2+^ signal is an important regulator of endothelial permeability, particularly in lung edema [141], it is not currently known whether CalDAG-GEFs are involved or how Rap1 signaling may intersect with Ca^2+^ signaling. Despite that, a recently uncovered role in nitric oxide (NO) release is where Rap1 and Ca^2+^ signaling intersect and this is one of the most physiologically impactful functions of Rap1 [11].

The endothelial cell-specific deletion of Rap1 during development leads to an impaired vascular barrier and is not compatible with vascular maturation [142]. However, the deletion of both Rap1 isoforms after birth does not lead to increased vascular permeability in most vascular beds. Instead, it leads to severely attenuated NO release and impaired vasodilation resulting in hypertension [143,144]. Underlying this defect is the impaired activation of endothelial nitric oxide synthase (eNOS), in part due to defective sensing of shear-stress of flowing blood, which is a major physiological regulator of NO release [143]. The shear stress-induced activation of eNOS, largely regulated by eNOS phosphorylation [145], is impaired in Rap1-deficient endothelial cells [143]. The studies of the underlying mechanism revealed that Rap1 promotes the assembly of the endothelial junctional mechanosensing complex (comprised of PECAM-1, VE-Cadherin, and VEGFR2) and is critical for VEGFR2 transactivation and signaling to NO release [143]. In addition to promoting phosphorylation-dependent eNOS activation, Rap1 is required for Ca^2+^-dependent eNOS activation, as endothelium-specific deletion of Rap1 leads to a significant impairment of acetylcholine induced-Ca^2+^-dependent vasodilation and is sufficient to induce hypertension in vivo [128,143]. Conversely, (Epac-induced) Rap1 activation induces a sustained increase in cytosolic Ca^2+^, eNOS activity, and subsequent NO production contributing to endothelium-dependent vasorelaxation [129] (Figure 5). While the underlying mechanisms are still under investigation, these findings underscore the fundamental role of cross-talk between Rap1 and Ca^2+^ signaling in EC homeostasis.

In summary, endothelial Rap1 signaling plays an important role in the fundamental endothelial processes regulated by Ca^2+^ signaling—the dynamic regulation of endothelial barrier and release of NO. While the identity of some of the Rap1 effectors and GEFs regulating these processes is known, more research is needed to understand the exact underlying mechanisms. Such analysis may reveal novel therapeutic targets for treating diseases associated with endothelial dysfunction such as atherosclerosis, myocardial infarction and hypertension.

## 11. Conclusions and Future Perspectives

In two different signaling modalities, as an upstream activator and downstream effector, active Rap1 intersects with Ca^2+^ signaling to control important functions in the multiple tissues. The significance of Rap1/Ca^2+^ cross-talk is particularly evident in the CNS, where increased Ca^2+^ from external Ca^2+^ influx or internal Ca^2+^ release activates Rap1 to stimulate B-Raf/ERK signaling. This, in turn, controls neuronal function and gene expression [77,79]. However, it is not known how these Ca^2+^ signals, with different frequencies and amplitudes, activate Rap1 to induce ERK activation. Better understanding of the kinetics of Rap1 activity and identity of Rap1 activating GEFs involved in Ca^2+^ signaling is needed. To this end, novel tools [146] and model organisms [143] might help provide answers.

Studies of Rap1 in the endothelium, heart and platelets have revealed its novel effectors and functions important for tissue and organ homeostasis [95,122,128,129,143]. In the endothelium, eNOS has been identified as a novel Rap1 effector [143]. eNOS activity is regulated by phosphorylation events, particularly in response to shear-stress, but also depends on Ca^2+^ signaling elicited downstream from several GPCR-agonists [147,148,149]. Furthermore, store-operated Ca^2+^ entry is required for sustained eNOS activation in endothelial cells [150]. Interestingly, via distinct mechanisms, Rap1 intersects with Ca^2+^ signaling in smooth muscle cells, promoting their relaxation. Thus, in both tissues, Rap1 controls vascular tone [128]. CalDAG-GEFIII, a potential Rap1 activator, has been implicated in hypertension by genome-wide association studies [53]. However, its function in the endothelium has not been widely studied and has been reported only once to date [54].

It is becoming evident that Rap1 signaling is essential for the function of multiple organs, and that some key Rap1 functions rely on cross-talk with Ca^2+^ signals. A thorough characterization of Rap1 regulators, effectors, and the mechanisms connecting them in time and space may provide novel therapeutic strategies for several cardiovascular and neurological diseases.

## Figures and Tables

**Figure 1 ijms-21-01616-f001:**
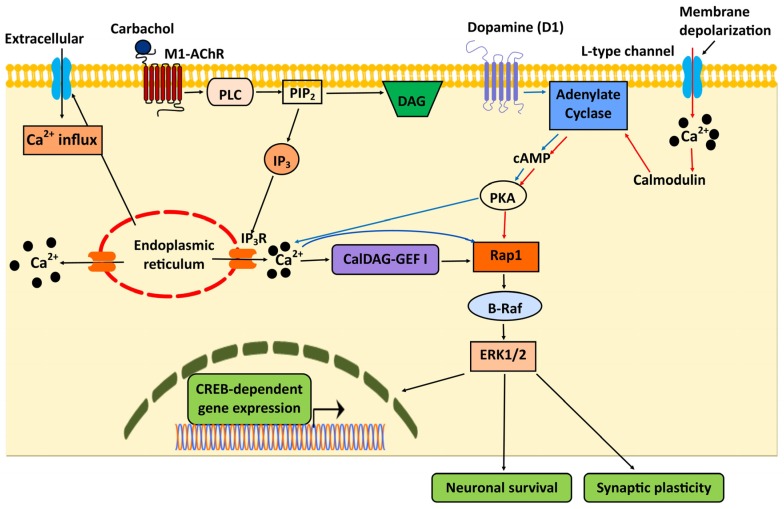
Rap1 and Ca^2+^ signaling in the central nervous system (CNS). Rap1/B-Raf/ERK is the main pathway controlling synaptic plasticity, gene expression and neuronal survival, and is induced by Ca^2+^ signal upstream from Rap1 activation. Muscarinic acetylcholine receptor (M1-AChR)-induced Ca^2+^ release and extracellular Ca^2+^ influx activate Rap1 via CalDAG-GEFI. Dopamine (D1)-induced cAMP/PKA activation followed by Ca^2+^ release from intracellular stores (blue arrows), results in CREB phosphorylation and gene expression. Membrane depolarization and Ca^2+^ influx via L-type Ca^2+^ channels activates cAMP/PKA/Rap1/B-Raf via calmodulin (red arrows), modulating synaptic plasticity and neuronal survival.

**Figure 2 ijms-21-01616-f002:**
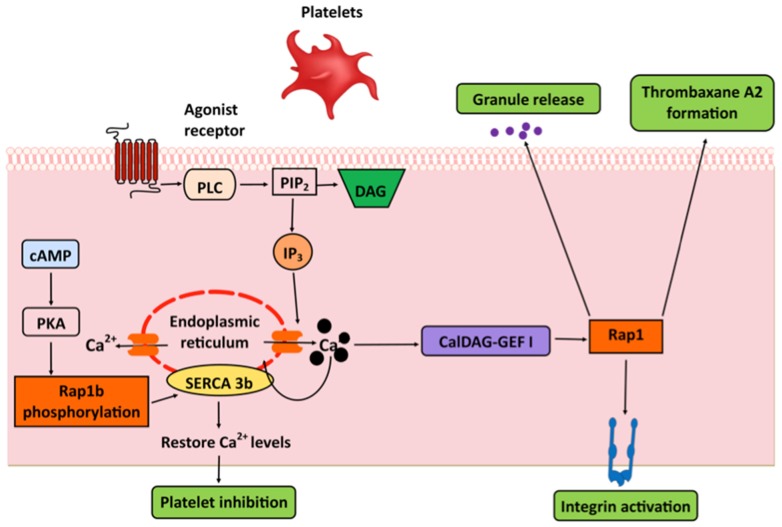
Rap1 and Ca^2+^ cross-talk in platelet function. Downstream from agonist receptors, CalDAG-GEFI links the intracellular Ca^2+^ rise with Rap1 activation, promoting integrin activation, thromboxane A2 formation and granule release. Rap1b physical interaction with sarco/endoplasmic reticulum Ca^2+^ ATPase (SERCA 3b), regulated by phosphorylation, modulates Ca^2+^ re-uptake and platelet activation.

**Figure 3 ijms-21-01616-f003:**
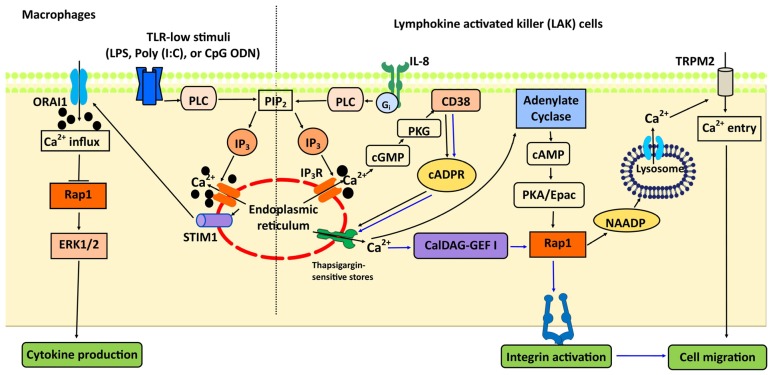
Rap1 and Ca^2+^-cross-talk in immune cells. Ca^2+^ signal upstream from Rap1 either inhibits or activates it, eliciting different responses in immune system. In macrophages (to the left of the dotted line), low doses of TLR agonists trigger signaling which induces Ca^2+^ release from ER and oligomerization of stromal interaction molecule 1 (STIM1), leading to the opening of the plasma membrane Ca^2+^ release-activated Ca^2+^ channel protein (ORAI1) channels and Ca^2+^ influx, which subsequently causes Rap1 inhibition and ERK activation to induce cytokine production. In lymphokine activated killer (LAK) cells (to the right of the dotted line), CD38 activation, downstream from the IL8 receptor, induces cyclic ADP-ribose (cADPR) production, enhancing Ca^2+^ release from thapsigargin-sensitive stores (blue lines), which promotes the activation of adenylyl cyclase. Downstream, cAMP-activated Epac/Rap1 induce nicotinic acid adenine dinucleotide phosphate (NAADP) production, resulting in CICR through TRPM2 channel, promoting cell migration.

**Figure 4 ijms-21-01616-f004:**
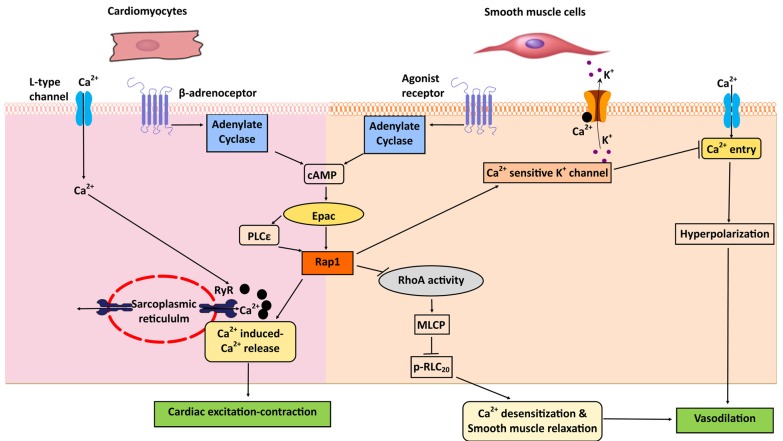
Rap1 and Ca^2+^ signaling in cardiac and smooth muscle function. In cardiomyocytes, cAMP/Epac-activated Rap1 stimulates Ca^2+^-induced Ca^2+^ release (CICR) from sarcoplasmic reticulum stores through ryanodine receptors (RyR) to regulate excitation–contraction coupling. In smooth muscle cells, Rap1 inhibits RhoA activity and relieves the disinhibition of myosin light-chain phosphatase (MLCP), which decreases myosin regulatory light chain of myosin (RLC_20_) phosphorylation, promoting Ca^2+^ desensitization and smooth muscle relaxation. Furthermore, Rap1 activation induces smooth muscle hyperpolarization by decreasing Ca^2+^ entry through opening of endothelial Ca^2+^-sensitive K^+^ channel to promote vasodilation.

**Figure 5 ijms-21-01616-f005:**
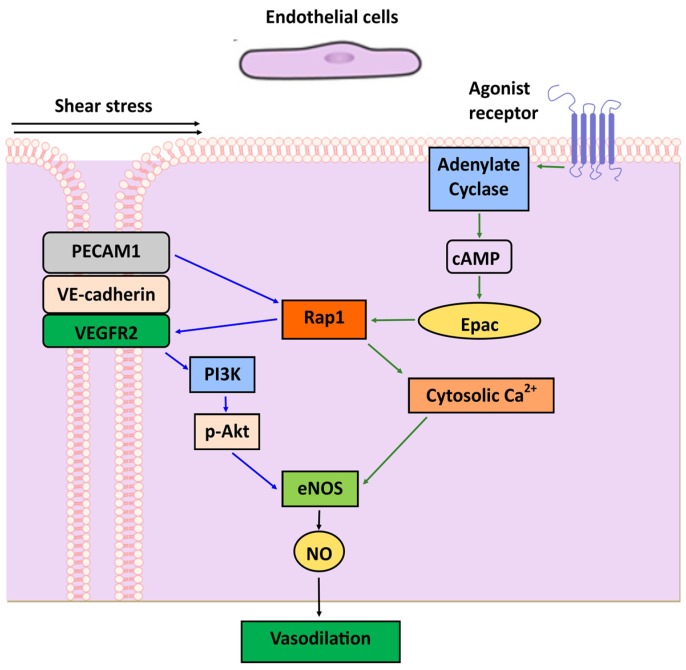
**Role of Rap1 in endothelial cell function.** Rap1 promotes nitric oxide (NO) and endothelial function in response to Ca^2+^-dependent agonists and by sensing shear stress and promoting signaling to eNOS phosphorylation downstream from the endothelial junctional mechanosensing complex consisting of PECAM-1, VE-cadherin and VEGFR2. Upon cAMP increase, Epac-activated Rap1 induces a sustained increase in cytosolic Ca^2+^, eNOS activity and subsequent NO production and vasodilation.

**Table 1 ijms-21-01616-t001:** Domain structure and specificity of Rap1 regulatory proteins.

Gene Symbol	Name of the Protein	Molecular Structure Domains	Protein Length	GEF Activity
*RAPGEF3*	Epac1	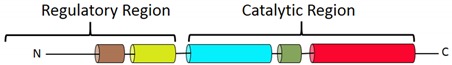	881	Rap1, Rap2
*RAPGEF4*	Epac2	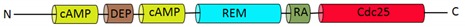	1011	Rap2
*RASGRP2*	CalDAG GEF-I	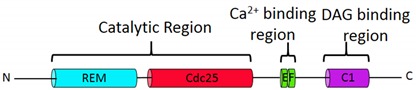	609	Rap1a>N-Ras
*RASGRP*	CalDAG GEF-II	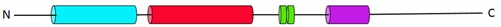	797	H-Ras, R-Ras
*RASGRP3*	CalDAG GEF-III	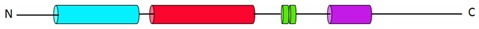	689	H-Ras, R-Ras, M-Ras, Rap1a, Rap2a
*RASGRP4*	CalDAG GEF-IV	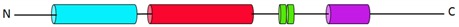	673	H-Ras

REM: Ras-exchange motif; Cdc25: catalytic Cdc25 homology domain; EF:Ca2+ binding EF hand; C1: Diacylglycerol binding motif; cAMP: cyclic adenosine monophosphate binding domain; DEP:Dishevelled, Egl-10, Pleckstrin region; RA: Ras association domain.

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
