# Peer review of "Integration of Rap1 and Calcium Signaling"

_ijms, 2020, doi:10.3390/ijms21051616_

Round 1

Reviewer 1 Report

The authors provided a good review about the contribution of Rap-1 in calcium signaling. However, there are several issues they need to focus on. Please see my comments.

Major Comments:

  1. Although the text is written well but to me it lacks the coherency. Most sentences read well but they are not connected together. For example paragraph starting line 61.
  2. Text contains several grammatical errors and needs extensive English editing.

Minor Comments:

  1. Line 29, close the parenthesis.
  2. Lines 46-48, the sentence “Furthermore….” Is vague and needs to be rewritten for clarity.
  3. Line 79, change “,” to “.”.
  4. Lines 92-93, sentence “Epac1…” is incomplete and does not make sense; please rewrite it.
  5. Line 108, remove “.” After channels
  6. Lines 130-132, sentence “Among those…” needs revision.
  7. The paragraph in lines 134-141 is very complex and needs grammatical correction in many places; please revise it.
  8. Line 153, remove and after “PIP3”.
  9. Lines 165-167, sentence “While….. ” is grammatically incorrect.
  10. Lines 174, what is the definition of authors from peripheral tissues? IS heart considered as a central organ? I am completely confused!

Author Response

We thank the Reviewer for their complimentary comments on the quality of our review.  Our point-by-point responses are listed below (in bold font).

Major Comments:

  1. Although the text is written well but to me it lacks the coherency. Most sentences read well but they are not connected together. For example paragraph starting line 61.

Response: We thank the Reviewer for finding the text well written. We have reviewed the manuscript for coherency and edited connecting sentences, as needed. All changes are visible with “Track changes” option “on”. Specifically, we revised the paragraph starting on line 61, as suggested.

 Text contains several grammatical errors and needs extensive English editing.

Response: We have carefully reviewed the manuscript for grammar and syntax. Ms. Shana Maker (see Acknowledgements), who is a native speaker of (American) English, provided invaluable assistance in that regard.

Minor Comments:

  1. Line 29, close the parenthesis.

Response: the parenthesis on line 29 was closed.

  1. Lines 46-48, the sentence “Furthermore….” Is vague and needs to be rewritten for clarity.

Response: the sentence on lines 46-48 was rewritten.

  1. Line 79, change “,” to “.”.

Response: “,” was changed to “.” (now line 98).

  1. Lines 92-93, sentence “Epac1…” is incomplete and does not make sense; please rewrite it.

Response: the sentence “Epac1…” (now lines 111-112) was rewritten.

  1. Line 108, remove “.” After channels

Response: “.” after channels was removed (now line 127).

  1. Lines 130-132, sentence “Among those…” needs revision.

Response: The sentence “Among those…” (now lines 153-155) was revised.

  1. The paragraph in lines 134-141 is very complex and needs grammatical correction in many places; please revise it.

Response: The paragraph (now lines 157-165) was revised for clarity.

  1. Line 153, remove and after “PIP3”.

Response: “and” after “PIP3” was removed (now line 177), and the following sentence rewritten for clarity.

  1. Lines 165-167, sentence “While….. ” is grammatically incorrect.

Response: the sentence, now on lines 189-191, was rewritten.

  1. Lines 174, what is the definition of authors from peripheral tissues? IS heart considered as a central organ? I am completely confused!

Response: "peripheral vasculature" is referred to vasculature outside of the brain and heart. We revised the sentence (now on lines 197-199) for clarity.

Reviewer 2 Report

The manuscript is well written and cover an interesting area of research.

In Heart: excitation-contraction coupling; cardiac hypertrophy the author should discuss the role of the Rap1 in myocardial ischemia/reperfusion injury 

The reference list need to be improved citing for instance the role of rap1 in ischemia/reperfusion in hearth.

see: Deletion of Rap1 protects against myocardial ischemia/reperfusion injury through suppressing cell apoptosis via activation of STAT3 signaling.

Cai Y, Ying F, Liu H, Ge L, Song E, Wang L, Zhang D, Hoi Ching Tang E, Xia Z, Irwin MG.

FASEB J. 2020 Feb 5. doi: 10.1096/fj.201901592RR. [Epub ahead of print]

Author Response

We thank the Reviewer for finding our manuscript well written and covering an interesting area of research.

“In Heart: excitation-contraction coupling; cardiac hypertrophy the author should discuss the role of the Rap1 in myocardial ischemia/reperfusion injury 

The reference list need to be improved citing for instance the role of rap1 in ischemia/reperfusion in hearth.

see: Deletion of Rap1 protects against myocardial ischemia/reperfusion injury through suppressing cell apoptosis via activation of STAT3 signaling.

Cai Y, Ying F, Liu H, Ge L, Song E, Wang L, Zhang D, Hoi Ching Tang E, Xia Z, Irwin MG.

FASEB J. 2020 Feb 5. doi: 10.1096/fj.201901592RR. [Epub ahead of print]”

Response: While we completely agree with the Reviewer that the function of (small GTPase) Rap1 in myocardial ischemia/reperfusion injury would be relevant in the chapter on Heart, we have not found any reports on that topic.  

Unfortunately, the paper the Reviewer is referring to, pertains to a different molecule with the same acronym: RAP1. The Cai et al. paper pertains to Repressor-activator protein 1 (RAP1), a telomere-associated protein, and, thus, is not relevant to our review on a small GTPase Rap1. As a side note, one of us (Dr. Kosuru) has previously published an article on the telomere-associated protein RAP1 (Cell Cycle 16:1765), and thus is very familiar with that protein.

Round 2

Reviewer 1 Report

There are still a few punctuation errors.

Line 13, a "," is needed after "inactive states".

Line 68, please remove an unnecessary space after "[16, 17]".

Line 79-80, the sentence "This inhibits...." to me needs revision. I suggest "This signaling inhibits Rap1 prenylation and its membrane localization resulting in cell scattering".

Line 91-92, the sentence "In addition... " is similarly not correct to me. I suggest such a sentence, "In addition to localizing at the plasma membrane, Rap1 is present at other cellular membranes..."

lines 95-96, I am not sure what the authors wanted to say in the sentence "Epac1..."; elaboration on that sentence is recommended.

Lines 132-136, the sentence "Rap1.." still needs correction. I suggest separate it into two sentences.

Author Response

Response: we thank the Reviewer for carefully reading our manuscript. We have corrected the punctuation errors throughout the manuscript, and the errors detailed below.

Line 13, a "," is needed after "inactive states".

Response: a “,” was inserted after “inactive states”.

Line 68, please remove an unnecessary space after "[16, 17]".

Response: an unnecessary space after "[16, 17]" was removed (now line 66).

Line 79-80, the sentence "This inhibits...." to me needs revision. I suggest "This signaling inhibits Rap1 prenylation and its membrane localization resulting in cell scattering".

Response: Reviewer’s suggestion was considered and the sentence (now on lines 74,75) was revised.

Line 91-92, the sentence "In addition... " is similarly not correct to me. I suggest such a sentence, "In addition to localizing at the plasma membrane, Rap1 is present at other cellular membranes..."

Response: The sentence (now on lines 106-107) was revised as suggested.

lines 95-96, I am not sure what the authors wanted to say in the sentence "Epac1..."; elaboration on that sentence is recommended.

Response: As suggested, we elaborated on the sentence (now on lines 110-112) to better explain the concept.

Lines 132-136, the sentence "Rap1.." still needs correction. I suggest separate it into two sentences.

Response: The sentence, (now on lines 154-157) was separated into two sentences, as suggested.